# Prevalence and Determinants of Changes in Physical Activity and Sedentary Behavior during and after the COVID-19 Pandemic: A Swedish Repeated Cross-Sectional Study

**DOI:** 10.3390/ijerph21080960

**Published:** 2024-07-23

**Authors:** Birgitta Kerstis, Maria Elvén, Kent W. Nilsson, Petra von Heideken Wågert, Jonas Stier, Micael Dahlen, Daniel Lindberg

**Affiliations:** 1Division of Caring Sciences, School of Health, Care and Social Welfare, Mälardalen University, 72123 Västerås, Sweden; 2Division of Physiotherapy, School of Health, Care and Social Welfare, Mälardalen University, 72123 Västerås, Sweden; maria.elven@mdu.se (M.E.); petra.heideken.wagert@mdu.se (P.v.H.W.); 3Division of Public Health, Care and Social Welfare, Mälardalen University, 72123 Västerås, Sweden; kent.nilsson@regionvastmanland.se; 4Center for Clinical Research, Central Hospital of Västerås, Uppsala University, 75310 Uppsala, Sweden; 5Division of Social Work, School of Health, Care and Social Welfare, Mälardalen University, 72123 Västerås, Sweden; jonas.stier@mdu.se; 6Department of Marketing and Strategy, Stockholm School of Economics, 11383 Stockholm, Sweden; 7Department of Social Work, School of Behavioural, Social and Legal Sciences, Division of Social Work, Örebro University, 70182 Örebro, Sweden; daniel.lindberg@oru.se

**Keywords:** physical activity, sedentary behavior, COVID-19, changes, global crisis

## Abstract

Physical activity (PA) and sedentary behavior (SB) changed during the COVID-19 pandemic; hence, this study examined PA and SB at four time points between December 2019 and December 2022. The participants’ PA decreased during the pandemic and did not recover afterwards. Among women, PA increased slightly in 2022 but not at all in men. From 2019 to 2020, SB increased and then decreased to near the pre-pandemic level in both sexes. Regarding age, PA decreased in the oldest age group (65–79 years) across all time points, while SB increased in all age groups during 2019–2020 and then returned close to pre-pandemic levels among the two middle age groups (30–64 years), but not among the youngest and oldest groups. Considering occupation, PA decreased from 2020 to December 2022 among retired and “other” participants, while SB decreased among nonmanual workers and retired participants. The regression models associated better self-reported health, male sex, and those born overseas with higher PA. Higher age, better self-reported health, poor education, and later survey time points were associated with lower SB. These findings highlight the need to return PA and SB to at least pre-pandemic levels and that subgroups may need different interventions.

## 1. Introduction

Physical activity (PA) refers to any bodily movement [1], while sedentary behavior (SB) refers to time spent sitting or lying while awake with low energy expenditure in the context of transportation, occupational, educational, home, and community settings [2]. PA has significant health benefits for the heart, body, and mind [1]. Regular PA can prevent and manage noncommunicable diseases, such as heart disease, stroke, diabetes, and several cancers [3]. In contrast, SB is positively associated with an elevated risk for all-cause mortality, cardiovascular diseases, cancer, hypertension, depression, and metabolic disorders, such as diabetes mellitus [4]. Accordingly, an umbrella review of 40 systematic reviews showed that a reduction in SB could contribute to the prevention of chronic diseases [5]. However, the World Health Organization’s (WHO) Global Action Plan on Physical Activity 2018–2030 urges people to increase their levels of PA and decrease their SB to meet the global target of a 15% relative reduction in the prevalence of insufficient PA by 2030 [6].

In December 2019, a novel coronavirus (initially named nCoV) emerged in China and quickly spread worldwide, prompting the WHO to declare a global pandemic on 11 March 2020. By 5 May 2023, the WHO determined that the ongoing Coronavirus Disease 2019 (COVID-19) pandemic was no longer a public health emergency of international concern, although the pandemic itself continued. By March 2023, the WHO had accounted for 760,360,956 confirmed cases of COVID-19, including 6,873,477 deaths [7].

Reviews have described a decrease in PA and an increase in SB during the pandemic [8,9]. It was observed in another review that PA decreased in all age groups, independent of sex, in the majority of the included studies (32 of 57 studies reviewed), while 5 studies described a significant increase in PA [10]. An article from the US noted that many organizations embraced flexible working practices during the pandemic and moved away from physical offices to take advantage of the benefits of the flexibility afforded by working from home [11], which in turn may have affected the possibilities for PA and the level of SB. A German study describes that mobility patterns during the nationwide restrictions in the spring of 2020 shifted towards inactivity or low-intensity activities, potentially leading to considerable and lasting health risks [12]. Concerning whether PA and SB had returned to pre-pandemic levels by 2023, a Saudi Arabian scoping review including 19 studies and an Australian study described decreased PA and increased SB during and following the COVID-19 lockdowns, which had not yet returned to pre-pandemic levels [13,14].

Further, we can speculate that life satisfaction deteriorated during the pandemic because it is associated with PA and SB levels [15]. In addition, PA is continuously promoted to support healthier and happier lives [16] with higher life satisfaction [17].

According to one review, there was an initial increase in depression and anxiety during the pandemic, while changes in mental health symptoms varied substantially across studies, suggesting that different populations responded differently to the pandemic and related containment measures [18]. In a combined systematic review and meta-analysis, incremental increases in body weight were described as an alarming effect of lockdowns during the COVID-19 pandemic, which led to a potentially higher incidence of overweight, obesity, and related health risks [19]. Hence, it becomes important to study people’s PA and SB behaviors during and after a crisis, such as the COVID-19 pandemic, to be able to manage the possible emergence of new pandemics.

This study focuses on the Swedish population’s levels of PA and SB before, during, and after the COVID-19 pandemic. The Swedish pandemic strategy differed from other countries’ more restrictive measures [20], as it favored voluntary compliance with the government’s recommendations [21]. In Sweden, most measures against COVID-19 ended in February 2022 [22]; therefore, the term “postpandemic” is used in the present study for the survey conducted in December 2022. In Sweden, the proportion of people reporting that they exercise regularly has increased since 1980 by up to 60% in 2020. In the 16–84-year-old age group, reporting being physically active for at least 30 min per day was unchanged at 65% over the last decade [23]. During the pandemic, a less favorable development in PA was observed among Swedish women compared with men and among blue-collar (i.e., manual) compared to white-collar (i.e., nonmanual) workers [24]. Another Swedish study indicated increased social inequalities in PA across age, education, occupation, income, and place of birth, but not regarding sex, during the pandemic compared to before the pandemic [25].

Further, the present research group has published data from two earlier surveys [26,27] showing that PA decreased and SB increased during the pandemic, but recovered in January 2022, although not to pre-pandemic levels. The present study is a follow-up of these two earlier studies and contributes to the literature on what happened to PA and SB in Sweden before, during, and after the pandemic, and after the pandemic is an additional element in this paper compared to those previously published [26,27]. It is also valuable to measure whether life satisfaction and self-reported health were associated with PA and SB, as people’s physical and mental health are affected by their PA and SB levels [28,29,30,31]. 

We hypothesized that PA decreased and SB increased during the pandemic and then recovered postpandemic.We hypothesized that changes in PA and SB during and after the pandemic were associated with age, confirmed COVID-19 infection, education, life satisfaction, occupation, place of birth, self-reported health, and sex.

Therefore, this study examined PA and SB at four time points in a Swedish population.

## 2. Subjects and Methods

### 2.1. Recruitment of Participants and Data Collection

This population-based study with a repeated cross-sectional design included participants aged 18–79 years who were recruited through Novus, a Swedish survey management service, including 50,000 potential participants. The Novus data are a population-based randomly recruited Swedish panel that provides representative results. This study was cross-sectional as the participants were anonymized and not possible to retrace. The data were collected from responses to web-based questionnaires. The first survey was emailed to 2000 people on 7 December 2020. A follow-up email was sent on 11 December 2020. Of these, 1035 participants (52%) anonymously completed the survey. In the 2020 survey, the participants were also asked to retrospectively self-report their PA and SB during a normal week in December 2019, which was 3 months before the WHO declared a worldwide COVID-19 pandemic. The second survey was emailed to 1894 new individuals on 21 January 2022, with a follow-up email on 30 January 2022, which was answered by 1095 participants (55%). The third survey was emailed to 1880 new individuals on 7 December 2022, with a follow-up email on 11 December 2022, which was answered by 1027 participants (55%). All participants provided their informed consent to take part in the survey when answering the questionnaire. This study was conducted according to the 2013 Declaration of Helsinki [32] and Swedish law [33]. After the material was collected, Novus deleted all personal data connections, which were not made accessible to the researchers in the present study.

### 2.2. Collection of Data

The surveys included questions about sociodemographic factors, PA, SB, confirmed COVID-19 infection, life satisfaction, and self-reported health. The sociodemographic questions included age, cohabitation status, highest education, occupation (in the analyses, for example, occupation was recategorized as parental leave, long-term sick leave, early retired, unemployed, and other) place of birth, and sex. The International Physical Activity Questionnaire—Short Form (IPAQ-SF) was used to record the duration of PA and SB during the last 7 days, including aerobics, cycling, gardening, running, and walking as a proxy for PA, and sitting as a proxy for SB [34]. To avoid outliers in this study, the participants who self-scored a PA with a minimum 10 min and a maximum 960 min per day were included, while the remaining participants were excluded (*n* = 196, *n* = 8, respectively). We also excluded those with higher SB than 960 min per day (*n* = 31).

The question, “In general, would you like to say that your health is?” was answered using a 5-point Likert scale (1 = poor to 5 = excellent). The question, “How satisfied are you with life for the moment?” was measured on a 10-point Likert scale, where a higher score indicated higher life satisfaction. Also included in the last three surveys was a question regarding confirmed COVID-19 infection (Yes/No).

### 2.3. Analyses

Data were analyzed using descriptive statistics: frequencies and percentages for categorical variables, and medians, means, and standard deviations (SD) for ordinal and continuous variables. Due to the skewed distribution of PA, we used the Kruskal–Wallis and Mann–Whitney U tests for time-based comparisons. For SB, which was normally distributed, ANOVA and Student *t*-tests were used. Spearman’s rho was applied for correlations to account for scale differences between PA and SB. Multiple regression analysis investigated associations between independent variables (PA and SB) and covariables. Education replaced occupation in the regression due to more reliable recall. Covariables included age, place of birth, confirmed COVID-19 infection, education, life satisfaction, self-reported health, sex, and survey time points (December 2020, January 2022, December 2022). A log10 transformation was applied to PA due to positive skewness [35]. Interaction terms were excluded from the final regression as no significant interactions were found between PA and survey time points. Models were built based on correlation coefficients to avoid multicollinearity [36], first entering the variable with the strongest bivariate correlation. Variables not correlated with PA (age, COVID-19 infection) or SB (place of birth, COVID-19 infection, sex) were excluded. We computed Cohen’s d using the standardized coefficient (β) and R square to measure the effect size of each predictor on the outcome variable (small effect: d ≈ 0.2, medium effect: d ≈ 0.5, and large effect: d ≈ 0.8). All tests were two-tailed with a significance level of *p* ≤ 0.05. Analyses were conducted using IBM SPSS Statistics (version 28.0).

## 3. Results

### 3.1. Description of the Sample

In 2020, the participants’ mean age was 50.6 years (SD 16.5), with 49.5% women (men, mean age 51.8, SD 16.0 versus women, mean age 49.4 SD 17.2). In January 2022, the participants’ mean age was 52.2 years (SD 16.5) with 51.1% women (men, mean age 52.4, SD 16.8 versus women, mean age 52.0 SD 16.6), while in December 2022, the participants’ mean age was 52.2 years (SD 16.7) with 51.1% women (men, mean age 52.5, SD 16.2 versus women, mean age 51.8 SD 17.2). Total life satisfaction increased from 6.5 in 2020 to 6.8 in January 2022 and 7.1 in December 2022 (*p* < 0.001). The confirmed COVID-19 infections increased from 5.3% in 2020 to 22.3% in January 2022 and 50% in December 2022 (Table 1). The correlation between self-reported health and life satisfaction was 0.473 (*p* < 0.001).

### 3.2. PA and SB in Relation to Sex

In total, PA decreased from 15.2 h per week in 2019 to 11.5 h per week in December 2022 (Table 2). Women’s PA decreased over the same period from 14.5 to 11.1 h per week, while men’s PA decreased from 15.9 to 11.9 h per week. However, the differences in men’s PA were not significant. PA was lower for women than men at all time points but was significantly lower only in January 2022. Overall, SB exhibited more significant changes over time compared to PA. SB initially increased during the pandemic between 2019 and December 2020 but decreased to the earlier level in December 2022. Concerning SB, men and women showed comparable development over time. During the pandemic, SB increased and returned close to the pre-pandemic level for men, while women’s SB increased and decreased slightly below the pre-pandemic level by December 2022. No differences in SB between men and women were found at the various measurement points in 2019, 2020, January 2022, and December 2022 (Table 2).

### 3.3. PA and SB in Relation to Age

The largest and only statistically significant decrease over time in PA occurred in the 65–79-year-old age group (from 19.2 h per week in 2019 to 9.4 h per week in December 2022). There were differences in PA between the age groups such that the oldest age group reported the most PA in 2019 and 2020, and the least PA in December 2022.

Regarding SB, the overall changes were statistically significant in all age groups. SB increased from 2019 to 2020 and decreased thereafter by varying amounts in the different age groups. Compared to the pre-pandemic (i.e., 2019) levels, the postpandemic (i.e., December 2022) levels were higher in the youngest and oldest age groups and approximately the same in the two middle-aged groups (Table 2).

### 3.4. PA and SB in Relation to Education and Occupation

PA decreased over time among those with primary and secondary education (compulsory or senior high school education), while no such change was observed among those with university education. No statistically significant differences in PA were observed between the education groups at any survey time point.

SB increased over time among those with compulsory school education and decreased among those with senior high school and university education. In general, higher education was associated with higher levels of SB at each survey time point. There were significant differences in both PA and SB between occupation groups at each survey time point. PA decreased significantly over time among retired and “other” participants. Nonmanual workers reported less PA than did the other groups in 2020, while due to a decrease among students (nonsignificant), retired, and “other”, these three groups ended with PA as low as nonmanual workers did in December 2022.

SB decreased in all occupational groups over time, but the differences between the surveys were significant only among nonmanual workers and retired participants. There were statistically significant differences between occupation groups, where students and nonmanual workers reported more SB than the other groups at all survey time points (Table 2).

### 3.5. PA and SB Associations with Covariables

The covariables in the regression model explained 4.3% of the total variance of PA (Table 3). Better self-reported health, lower education, male sex, and born overseas were associated with higher PA. Life satisfaction was not significant in the regression model but did correlate with PA (0.084, *p* < 0.001) and was therefore included in the model. The same logic applied to education, with a correlation of −0.058 (*p* < 0.001). Age, confirmed COVID-19 infection, and survey time points did not significantly correlate with PA and were therefore excluded.

The covariables in the regression model explained 8.9% of the total variance in SB (Table 4). Higher age, better self-reported health, higher life satisfaction, lower education, and later survey time points were associated with decreased SB. Confirmed COVID-19 infection, place of birth, and sex did not significantly correlate with SB and were therefore excluded.

## 4. Discussion

Our main findings were that PA decreased during the pandemic and did not recover to pre-pandemic levels, which is in accordance with other studies [13,14]. In total, PA decreased in the surveys for women, the 65–79-year-old age group, and for retired and “other” occupational groups. The reviews have described that PA decreased and SB increased in many of the included studies during the pandemic [10,37]. In contrast, a large data analysis of high-income countries found that PA increased, and SB decreased [38]. In a Swedish study, PA increased between 2018 and 2021 [25]. In our study, we found significant differences between the age groups in 2019 and 2020, but not in 2022, as the oldest age group showed a remarkable decline in their PA but reported their highest level of PA in 2019 and 2020 with the lowest PA in 2022. Interestingly, this suggests that the initially identified difference in PA between age groups appeared to decrease over time. The decline in PA may have a more profound impact on the oldest age group since this group was described as having more vulnerable mental and physical health than the other groups [39].

Another main finding was that SB showed more pronounced variations over time than did PA in the present study. The previous studies showed that SB did not return to pre-pandemic levels [13,14], which was partly confirmed in some groups in our study. Overall, SB increased between 2019 and 2020 but subsequently declined by January 2022 and declined further by December 2022. The total SB in December 2022 was slightly below the reported levels in 2019. Men reported nearly identical levels of SB before and after the pandemic. In contrast, women reported slightly lower SB after the pandemic than before the pandemic. Our finding was in accordance with other studies describing that increased SB and higher education are associated [40,41]. It is important to promote decreased SB as this can contribute to increased PA [42], prevent chronic diseases [5], and promote public health [4].

We found that better self-reported health, lower education, male sex, and born overseas were associated with higher PA, while higher age, better self-reported health, lower education, and later survey time points were associated with decreased SB. There was a correlation between decreased PA and lower life satisfaction, which was in accordance with a Swedish study [17]. However, the correlation coefficient was low and life satisfaction was not a significant determinant of PA. Previous studies have shown that the years of schooling reduce adult mortality [43] and in an Indian study, members of lower social classes died more often due to infections [44]. However, we have not examined mortality, but our results could be in contrast with those studies. Furthermore, we found no support for whether confirmed COVID-19 infection affected PA during and after the pandemic.

Another Swedish study described increased occupational inequality in PA during the pandemic [24]. The authors concluded that in contrast to their blue-collar counterparts, individuals in white-collar professions were afforded remote work options, which might result in reduced commuting durations and heightened schedule flexibility, conceivably allowing these workers additional time for engagement in PA. Flexible working practices [11] probably benefited home workers’ PA. In the present study, no significant changes in PA over time were observed among manual and nonmanual workers, while PA decreased significantly among the retired and “other” participants. In the Swedish context, the retired participants comprised people aged over 65 years and almost 17% of the total group was aged over 70 years. The Swedish response to the pandemic was to urge those aged 70 and over to avoid social contact during the pandemic, which was also relevant for the observed decline in the oldest age group that naturally included more retired individuals than in the other age groups. Hence, the surveyed decline in PA can partly be explained by the Swedish pandemic strategy, which was based on more voluntary restrictions compared to other countries [20]. Furthermore, we found no support for whether confirmed COVID-19 infection or sex affected the participants’ SB during and after the pandemic.

Our results showed that the association between SB and later survey time points were in line with the findings described in the reviews [8,9]. In addition, independent of time points, the SB levels were higher among participants who were younger, had higher education, and lower self-reported health. Concerning the participants’ occupations, there was a significant decline in SB between 2020 and December 2022 among nonmanual workers. The Swedish pandemic strategy, coupled with the country’s extensive welfare system, characterized by relatively equal income distribution and a well-developed health infrastructure [45], likely served as a buffer against the exacerbated social inequalities in PA and SB during the pandemic. These collective measures could mitigate the impacts of individual-level inequalities. This hypothesis was further supported by our study’s finding that there was an absence of clear socioeconomic patterns, as neither higher education nor being born in Sweden were significantly linked to high levels of PA or low levels of SB. The study’s findings provide additional insights into the factors that can influence PA and SB during crises such as the COVID-19 pandemic, which highlight the importance of societal structures in maintaining public health.

### Limitations and Strengths

Nonmanual workers and participants born in Sweden were over-represented compared to the overall population [46]. The over-representation of certain groups is because people who participate in Novus panels and respond to questionnaires have a higher socioeconomic status than those who do not participate in Novus panels and do not respond. Another limitation is the nonresponse in the three surveys (i.e., 48%, 45%, and 45%, respectively) which might affect the sample’s representativeness and the validity of the results. Another limitation was the self-report surveys, as the unreliability of human memory is well-known [47]. However, self-report surveys are proven as an uncomplicated way of collecting the opinions of many people and are established for collecting information on perceived health and life satisfaction. Another limitation was not including body weight in the analyses. Increments in body weight are described as an alarming effect of lockdown during the COVID-19 pandemic, leading to a potentially higher incidence of overweight, obesity, and related health risks [19]. A problem with Likert scales is that they may vary from one subject to another. A strength is their uniformity, which generates good control over the data collection and often saves time because they are usually quick to complete. In the first survey in 2020, the participants retrospectively self-reported their PA and SB 1 year before the current survey period; therefore, there was a further risk that the participants overestimated their level of PA and underestimated their level of SB when they estimated their pre-pandemic levels as compared to real-time self-reports. Another strength was that the questionnaires included validated instruments such as IPAQ-SF [34]. The questions about self-reported health [48] and quality of life [49] are used in several studies.

This study had a repeated cross-sectional design; therefore, it is suitable for analyzing time trends in PA and SB. However, it simultaneously does not allow for any conclusions about causality between the covariables and PA and SB. Finally, a potential limitation is that the current research took place within a Swedish setting and therefore may not transfer to other settings. Nevertheless, its contribution of knowledge specifically from the Swedish setting could be a major strength because the Swedish COVID-19 strategy differed from other countries, especially at the beginning of the pandemic.

## 5. Conclusions

Our findings highlighted the evolving patterns of PA and SB during and in the aftermath of the COVID-19 pandemic, such as the noticeable decline in PA during the pandemic and postpandemic with little recovery. Additionally, SB initially increased but later decreased postpandemic to close to or slightly below the pre-pandemic levels, which may illustrate changes in lifestyle patterns during and after the global health crisis. Considering the results of this study, it may be particularly interesting to investigate some subgroups further. For example, the 65–79-year-old age group showed the most noticeable decline in PA while the SB levels simultaneously increased. The observation that the youngest and oldest age groups (i.e., 18–29 and 65–79 years old, respectively) did not recover to pre-pandemic SB levels is troubling. The decline in PA may have a more profound impact on the oldest age group since their health and PA levels were closely related. Although the effect size between self-reported health and PA is small (Cohen’s d = 0.193), it indicates a consistent, positive relationship where individuals who report better health engage in slightly more PA. The effect size for age (Cohen’s d = −0.229), and self-reported health (Cohen’s d = −0.128) and SB indicate that as age increases and self-reported health decreases, SB increases. While this effect size is still small, it is stronger than the effect of self-reported health on PA. For all other indicators, the very small Cohen’s d values suggest that the effect of the predictor on the outcome variables is minimal. For the youngest age group, the increase in SB may not lead to direct health problems but may have negative health consequences in the long run. It is important that PA and SB levels at least recover to pre-pandemic levels for good public health. It is also important to decrease SB in general to promote public health. Additionally, monitoring PA and SB levels and trends is necessary because these data can help identify subpopulations for preventive interventions, assess strategies to promote health in general, and prepare for the next crisis.

## Figures and Tables

**Table 1 ijerph-21-00960-t001:** Characteristics of the participants in the three surveys (i.e., December 2020, January 2022, and December 2022).

	December 2020	January 2022	December 2022
	*n* (%)	*n* (%)	*n* (%)
Sex			
Men	523 (50)	536 (49)	525 (49)
Women	512 (50)	559 (51)	559 (51)
Age groups			
18–29 years	171 (16)	141 (13)	139 (14)
30–49 years	317 (31)	335 (31)	335 (31)
50–64 years	288 (28)	312 (28)	312 (28)
65–79 years	259 (25)	307 (28)	307 (28)
Cohabitation status			
Not living with a partner	321 (48)	315 (30)	316 (31)
Living with a partner	710 (62)	743 (70)	710 (69)
Place of birth			
Born in Sweden	926 (90)	984 (93)	958 (93)
Highest education			
Compulsory school (9 years)	66 (6)	57 (6)	60 (6)
Senior high school	419 (41)	415 (39)	364 (35)
University	544 (53)	586 (55)	601 (58)
Other	2 (<1)	2 (<1)	2 (<1)
Occupation			
Students	87 (8)	86 (8)	67 (6)
Manual workers	230 (22)	237 (22)	234 (23)
Nonmanual workers	334 (33)	334 (32)	333 (32)
Self-employed	62 (6)	53 (5)	54 (5)
Other ^a^	318 (30)	350 (33)	343 (34)
Confirmed COVID-19 cases	55 (5)	244 (22)	514 (50)
	Mean (SD)	Mean (SD)	Mean (SD)
Life satisfaction (1–10) *	6.5 (2.5)	6.8 (2.3)	7.1 (2.3)
Self-reported health (1–5) *	2.9 (0.98)	3.0 (1.0)	3.0 (1.0)

Note: * A higher score indicates better life satisfaction or self-reported health, respectively. ^a^ Parental leave, long-term sick leave, early retired, unemployed, and other.

**Table 2 ijerph-21-00960-t002:** Physical activity (PA) and sedentary behavior (SB) over time, and by sex, age groups, place of birth, education, and occupation.

	Physical Activity (Mean Hours SD per Week)					Sedentary Behavior (Mean Hours SD per Week)				
	2019 *	December 2020	January 2022	December 2022	*p*-Value	2019 *	December 2020	January 2022	December 2022	*p*-Value
Total	15.2 (18.1)	12.6 (15.3)	11.4 (12.8)	11.5 (13.0)	**<0.001** ^b^	48.1 (24.8)	53.8 (26.2)	49.5 (24.4)	47.7 (24.8)	**<0.001** ^d^
Sex										
Men	15.9 (16.5)	13.4 (15.6)	12.4 (13.3)	11.9 (13.7)	0.145 ^b^	47.9 (23.7)	53.2 (26.3)	49.0 (25.1)	48.3 (24.8)	**<0.001** ^d^
Women	14.5 (16.7)	11.7 (15.0)	10.4 (12.3)	11.1 (12.2)	0.036 ^b^	48.2 (23.2)	54.5 (26.0)	51.1 (23.8)	47.1 (24.9)	**<0.001** ^d^
*p*-value	0.087 ^a^	0.052 ^a^	**0.002** ^a^	0.667 ^a^		0.915 ^c^	0.774 ^c^	0.402 ^c^	0.507 ^c^	
Age groups										
18–29 years	15.7 (16.6)	12.5 (16.4)	11.5 (13.9)	12.1 (14.4)	0.175 ^b^	53.9	62.2 (28.2)	62.0 (25.1)	58.7 (26.0)	**0.015** ^d^
30–49 years	12.8 (15.1)	11.1 (14.6)	12.1 (14.6)	11.5 (13.7)	0.251 ^b^	51.7	57.2 (27.2)	51.6 (23.8)	49.1 (24.1)	**<0.001** ^d^
50–64 years	15.5 (15.7)	12.5 (14.2)	11.5 (13.3)	13.0 (14.0)	0.263 ^b^	47.6	53.0 (24.8)	51.0 (24.7)	47.0 (25.2)	**0.003** ^d^
65–79 years	19.2 (18.6)	14.6 (16.4)	10.6 (9.2)	9.6 (9.5)	**<0.001** ^b^	39.5	45.1 (22.1)	40.3 (21.1)	42.0 (22.6)	**0.014** ^d^
*p*-value	**<0.001** ^b^	**<0.001** ^b^	0.873 ^b^	0.478 ^b^		**<0.001** ^d^	**<0.001** ^d^	**<0.001** ^d^	**<0.001** ^d^	
Place of birth										
Sweden	15.5 (16.5)	13.4 (14.5)	12.6 (12.2)	13.1 (13.3)	**0.026** ^b^	48.4 (24.0)	53.8 (25.4)	49.3 (23.4)	47.0 (23.0)	**<0.001** ^d^
Born overseas	17.6 (17.8)	13.7 (14.5)	15.3 (15.3)	13.7 (10.1)	0.429 ^b^	43.1 (25.3)	51.7 (28.2)	49.6 (27.1)	43.3 (21.4)	0.095
*p*-value	0.273 ^a^	0.929 ^a^	0.289 ^a^	0.112		**0.010** ^c^	0.219 ^c^	0.842 ^c^	0.280 ^c^	
Education										
Compulsory school	-	14.2 (14.1)	9.1 (8.3)	8.9 (9.4)	**0.014** ^d^	-	43.6 (23.8)	45.3 (26.7)	45.5 (26.7)	**<0.001** ^d^
Senior high school	-	14.1 (18.1)	13.1 (15.5)	11.5 (14.1)	**0.002** ^d^	-	51.4 (26.9)	47.0 (24.7)	45.3 (27.1)	**0.003** ^d^
University	-	11.3 (12.8)	10.3 (10.6)	11.7 (12.6)	0.350	-	57.0 (25.4)	51.9 (23.8)	49.4 (23.1)	**0.041** ^d^
*p*-value	-	<0.393 ^b^	<0.198 ^b^	<0.058 ^b^		-	**<0.001** ^d^	**<0.003** ^d^	**<0.041** ^d^	
Occupation										
Student	-	11.6 (11.9)	12.0 (11.2)	9.6 (8.2)	0.632 ^b^	-	66.2 (24.5)	60.3 (25.7)	60.3 (25.0)	0.152 ^d^
Manual worker	-	16.3 (18.9)	14.7 (17.0)	16.9 (19.0)	0.672 ^b^	-	47.6 (26.7)	44.8 (23.1)	44.0 (24.3)	0.364 ^d^
Nonmanual worker	-	8.8 (9.1)	9.1 (9.9)	8.7 (9.6)	0.586 ^b^	-	62.1 (26.0)	58.3 (22.6)	53.0 (25.2)	**<0.001** ^d^
Self-employed	-	14.6 (17.2)	17.7 (16.6)	13.9 (15.2)	0.216 ^b^	-	50.2 (26.4)	43.6 (24.1)	46.0 (23.3)	0.407 ^d^
Retired	-	13.1 (13.6)	10.3 (9.0)	9.7 (9.8)	**0.014** ^b^	-	45.3 (22.0)	41.1 (21.6)	41.2 (22.3)	**0.004** ^d^
Other ^e^	-	16.0 (25.2)	11.6 (17.3)	9.0 (9.4)	**<0.001** ^b^	-	52.0 (23.2)	52.6 (27.6)	50.0 (26.3)	0.368 ^d^
*p*-value	-	**<0.001** ^b^	**<0.001** ^b^	**<0.001** ^b^		-	**<0.001** ^d^	**<0.001** ^d^	**<0.001** ^d^	

^a^ Mann–Whitney *U* test, ^b^ Kruskal–Wallis test, ^c^ Student’s *t*-test, ^d^ ANOVA, **bold** = statistically significant. ^e^ Parental leave, long-term sick leave, early retired, unemployed, other, - no data. * Recall: Asked in the 2020 survey.

**Table 3 ijerph-21-00960-t003:** Multiple regression analysis for PA associations with covariables.

Parameter	B (95% CI)	SE B	β	Cohen’s *d*	*p*-Value
Self-reported health	0.075 (0.060; 0.089)	0.007	0.189	0.193	<0.001
Life satisfaction	0.000 (−0.006; 0.005)	0.003	−0.003	−0.003	NS
Education	−0.044 (−0.065; −0.023)	0.011	−0.068	−0.069	<0.001
Sex (male vs. female)	−0.045 (−0.070; −0.020)	0.013	−0.058	−0.059	<0.001
Place of birth (Sweden vs. overseas)	0.049 (0.005; 0.093)	0.022	0.036	0.037	0.03

Abbreviations: B = unstandardized regression coefficient; 95% CI = 95% confidence interval; SE B = standard error; β = standardized beta coefficient; NS = nonsignificant. The variables are presented in the order they were added to the model, based on their correlation with PA.

**Table 4 ijerph-21-00960-t004:** Multiple regression analysis explaining SB with selected covariables.

Parameters	B (95% CI)	SE B	β	Cohen’s *d*	*p*-Value
Age	−0.319 (−0.362; −0.276)	0.022	−0.219	−0.229	<0.001
Self-reported health	−2.960 (−3.774: −2.146)	0.415	−0.122	−0.128	<0.001
Life satisfaction	−0.521 (−0.860; −0.183)	0.173	−0.052	−0.054	0.003
Education	4.921 (3.744; 6.098)	0.6	0.124	0.13	<0.001
Survey time points	−0.676 (−1.326; −0.026)	0.331	−0.031	−0.032	0.042

Abbreviations: B = unstandardized regression coefficient; 95% CI = 95% confidence interval; SE B = standard error; β = standardized beta coefficient; NS = nonsignificant. The variables are presented in the order they were added to the model based on their correlation with SB.

## Data Availability

The data presented in this study are openly available from FigShare at. https://doi.org/10.6084/m9.figshare.26134993.v1.

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
