# Peer review of "Prevalence and Determinants of Changes in Physical Activity and Sedentary Behavior during and after the COVID-19 Pandemic: A Swedish Repeated Cross-Sectional Study"

_ijerph, 2024, doi:10.3390/ijerph21080960_

Round 1

Reviewer 1 Report

Comments and Suggestions for Authors

Overall, the paper was well-written and well-thought-out.  I commend the authors for the diligent literature review highlighting the importance of increasing PA and decreasing SB as we continue through the post-pandemic era. 

With that, some suggestions are below.

Lines 59-62 should be reworded to highlight the association between life satisfaction and PA/SB levels. 

Lines 86-87. Can you authors provide more context to how the current study differs from the two previous studies mentioned [25,26].

Methods - possibly an explanation to why a cross-sectional study was used instead of a longitudinal study.  Specifically, explanation as to why the original cohort of survivors were not used as follow-up data.

Lines 266 - typo. From 2019 "to" 2020.

The paragraph starting on line 284 seems speculative.  Is there literature to support the thought that working remotely increases PA and decreases SB?  Is it possible that it could do the opposite  - in fact, sitting in front of a computer at home in place of walking around an office or place of employment could decrease PA? 

In limitations in line 318, the word "probably" should be deleted as it signifies that the author is speculating on these thoughts.

Author Response

Review report 1

Thank you, best review for your valuable comments. We look forward to your decision after our revisions following here.

Lines 59-62 should be reworded to highlight the association between life satisfaction and PA/SB levels.
Removed in line 63: known to be

Lines 86-87. Can you authors provide more context to how the current study differs from the two previous studies mentioned [25,26].
Added in line 93: after the pandemic is additional in this paper compared to those previously [26,27].

Methods - possibly an explanation to why a cross-sectional study was used instead of a longitudinal study.  Specifically, explanation as to why the original cohort of survivors were not used as follow-up data.
Added in line 113: The study was cross-sectional as the participants were anonymized and not possible to retrace.

Lines 266 - typo. From 2019 "to" 2020.
Thank you for the comment, changed in line 295: increased between 2019 and

The paragraph starting on line 284 seems speculative.  Is there literature to support the thought that working remotely increases PA and decreases SB?  Is it possible that it could do the opposite - in fact, sitting in front of a computer at home in place of walking around an office or place of employment could decrease PA?
Added in line 318: which might result in reduced commuting durations and heightened schedule flexibility, conceivably allowing these workers additional time for engagement in PA

In limitations in line 318, the word "probably" should be deleted as it signifies that the author is speculating on these thoughts.
Removed in line 351: probably

Reviewer 2 Report

Comments and Suggestions for Authors

The Editor   

Int J ER and Pub Health, ID: ijerph-3109382

Title: Changes in Physical Activity and Sedentary Behavior During and After the COVID-19 Pandemic: A Swedish Repeated Cross-Sectional Study By Birgitta Kerstis et al.

Thank you for asking me to review this paper.

1.Title should be: Prevalence and determinants of  changes in physical activity and sedentary behavior during and after the COVID-19 Pandemic:

2.Abstract.

Please give data in the abstract such as prevalence of PA and SB changes, as well as  OR and CI of the determinants, in the brackets.

3.Introduction.

There are many paras in this section along with half of the text that should be shifted to discussion. e g, following to be given under methods.

This study examined PA and SB at four time points in a Swedish population: December 2019 (assessed retro- spectively in 2020), January 2020, January 2022, and December 2022.

92
93
9

Hypothesis should be given in a line before the Objectives.

4.Subjects and Methods are better words because humans are not material

5. Recruitment of Participants.

Sample size calculation

Inclusion criteria

Exclusion criteria

Why did you stratified the sample

Did you validate the questionnaire, write a word

How many people not responded to emails?

Measures may be replaced by Collection of data

Please mention the problems of accuracy with Likert scale which may vary from one subject to another subject, under limitations.

Analysis by Statistical Analysis, which would be more clear to readers. This section is too much, need to be summarized or shifted to previous section.

6.Results.

Please give mean age for men and women separately in brackets, as COVID-19 is more common in men than women.

Why people of higher social class such as Professionals and business class were lowest?

Table 1, shows number and % of confirmed cases of COVID-19, do you mean all subjects had no COVID-19?

Table 1, last lines, please give SD along with means, for Life satisfaction and self reported health.

Table 2. Means are given without SD, please take opinion from Statistical Refree.

Tables 3, 4, Odds Ratio not given, to be consulted from Statistical refree.

Did you not ask body weight in the questionnaire? Which is important when studying PA and SB, it should be mentioned under limitations.

7.Discussion.

Do not write Aim and Hypothesis again under discussion, because these are parts of Introduction.

The aim was to examine PA and SB in a Swedish population before, during, and after

245

the pandemic at four time points: December 2019 (assessed retrospectively), January 2020,

246

January 2022, and December 2022. We hypothesized that the previously reported decrease

247

in PA and increase in SB during the pandemic [26,37] would have recovered to pre-

248

pandemic levels after the pandemic ended.

Write the most important finding of your study first then discuss by comparing with other studies.

Discuss about PA and SB among educated vs least educated people, during and after the disease.

Para 3, No need to mention, hypothesis again, go ahead with second most important finding. Discuss the role of Education as determinant of morbidity / Mortality

1.      IHME-CHAIN Collaborators. Effects of education on adult mortality: a global systematic review and meta-analysis. Lancet Public Health 2024; 9 (3): e155-e165.

2.      Singh RB, Singh V, Kulshrestha SK, Singh S, Gupta P, Kumar R, Krishna A, Srivastav SS, Gupta SB, Pella D, Cornelissen G. Social class and all-cause mortality in an urban population of North India. Acta Cardiol 2005; 60 (6): 611-617.

8.Limitations

Data were self reported by the patients, which is open to bias. The following sentence about participants with Univ. degree were overrepresented, is against your finding given in the Table 1, showing their low number.(Dec 2020, Jan 2022)

Nonmanual workers, participants with a university degree, and participants born in

316

Sweden were overrepresented compared to the overall population [45].

9.Give few references from 2023 and 2024

Comments on the Quality of English Language

Author Response

Dear Review 2,

Thank you for the opportunity to revise our manuscript in response to the reviewers’ comments. In the text below you will find our responses and a summary of the revisions made in our manuscript. The revisions done in the manuscript are marked with red colour. We thank you again for considering our manuscript and look forward to a positive decision.

Kind Regards,

Birgitta Kerstis
Associate professor
School of Health, Care and Social Welfare
Malardalen University
Vasteras, Sweden

Review 2

Thank you, best review for your valuable comments. We look forward to your decision after our revisions following here.

1.Title should be: Prevalence and determinants of changes in physical activity and sedentary behavior during and after the COVID-19 Pandemic:

Changed to Line 2: Prevalence and determinants of changes in physical activity and sedentary behavior during and after the COVID-19 Pandemic: A Swedish Repeated Cross-Sectional Study

2.Abstract.

Please give data in the abstract such as prevalence of PA and SB changes, as well as OR and CI of the determinants, in the brackets.
Answer: We are sorry but we cannot solve this due to the restriction of words in the abstract.

3.Introduction.

There are many paras in this section along with half of the text that should be shifted to discussion. e g, following to be given under methods.

This study examined PA and SB at four time points in a Swedish population: December 2019 (assessed retro- spectively in 2020), January 2020, January 2022, and December 2022.

92
93
9

Removed:
December 2019 (assessed retrospectively in 2020), January 2020, January 2022, and December 2022

Hypothesis should be given in a line before the Objectives.
Changed to: Hypothesis before the Objectives.

4.Subjects and Methods are better words because humans are not material.
Changed to in 107: Subjects and Methods 

  1. Recruitment of Participants.

Sample size calculation:

Answer: We did not perform power calculation but hope that effect sizes (Cohen`s d) and p-values provide a clear picture of the results. We hope that reporting these measures might be sufficient to understand the findings.

Inclusion criteria and Exclusion criteria

Answer: The Novus Sweden Panel consists of approximately 50,000 panellists. The panel is randomly recruited (one cannot sign up oneself to earn money or to influence public opinion) and is nationally representative in terms of age, gender, and region for individuals aged 18 to 79. Any biases in the panel structure are corrected by drawing a nationally representative sample from the panel.

Changed to line 111: The Novus data is a population-based randomly recruited Swedish panel that provides representative results.

Why did you stratified the sample
Removed: stratified

Did you validate the questionnaire, write a word:
Added in line 368: Another strength was that the questionnaires included validated instruments such as IPAQ-SF [35]. The questions about self-reported health [50] and quality of life [51] are used in several studies.

How many people not responded to emails?
Answer: We are sorry, but we have just the overall dropout.

Measures may be replaced by Collection of data
Replaced Measures in line 128: Collection of data

Please mention the problems of accuracy with Likert scale which may vary from one subject to another subject, under limitations.
Added Line 362: A problem with Likert scales is that they may vary from one subject to another.

Analysis by Statistical Analysis, which would be more clear to readers. This section is too much, need to be summarized or shifted to previous section.

Thank you for the comment, we agree and have summarized this section in line 145:
Data were analyzed using descriptive statistics: frequencies and percentages for categorical variables, and medians, means, and standard deviations (SD) for ordinal and continuous variables. Due to the skewed distribution of PA, we used the Kruskal-Wallis and Mann-Whitney U tests for time-based comparisons. For SB, which was normally distributed, ANOVA and Student t-tests were used. Spearman’s rho was applied for correlationx§s to account for scale differences between PA and SB. Multiple regression analysis investigated associations between independent variables (PA and SB) and covariables. Education replaced occupation in the regression due to more reliable recall. Covariables included age, place of birth, confirmed COVID-19 infection, education, life satisfaction, self-reported health, sex, and survey time points (December 2020, January 2022, December 2022). A log10 transformation was applied to PA due to positive skewness [36]. Interaction terms were excluded from the final regression as no significant interactions were found between PA and survey time points. Models were built based on correlation coefficients to avoid multicollinearity [37], first entering the variable with the strongest bivariate correlation. Variables not correlated with PA (age, COVID-19 infection) or SB (place of birth, COVID-19 infection, sex) were excluded. We computed Cohen’s d using the standardized coefficient (β) and R square to measure the effect size of each predictor on the outcome variable (small effect: d≈0.2, medium effect: d≈0.5, and large effect: d≈0.8). All tests were two-tailed with a significance level of p ≤ 0.05. Analyses were conducted using IBM SPSS Statistics (version 28.0).

6.Results.

Please give mean age for men and women separately in brackets, as COVID-19 is more common in men than women.
Added in line 194: with 49.5% women (men; mean age 51.8, SD 16.0 versus women; mean age 49.4 SD 17.2). In January 2022, the participants’ mean age was 52.2 years (SD 16.5) with 51.1% women (men; mean age 52.4, SD 16.8 versus women; mean age 52.0 SD 16.6), while in December 2022, the participants’ mean age was 52.2 years (SD 16.7) with 51.1% women (men; mean age 52.5, SD 16.2 versus women; mean age 51.8 SD 17.2).

Why people of higher social class such as Professionals and business class were lowest?

Clarified in line 318: The authors concluded that in contrast to their blue-collar counterparts, individuals in white-collar professions were afforded remote work options, which might result in reduced commuting durations and heightened schedule flexibility, conceivably allowing these workers additional time for engagement in PA.

Table 1, shows number and % of confirmed cases of COVID-19, do you mean all subjects had no COVID-19?
Clarified in line 142: Also included in the last three surveys was a question regarding confirmed COVID-19 infection (Yes/No).

Table 1, last lines, please give SD along with means, for Life satisfaction and self reported health.

Added Table 1: SD for Life satisfaction and self-reported health.

Table 2. Means are given without SD, please take opinion from Statistical Refree.

Added Table 2: SD for all means.

Tables 3, 4, Odds Ratio not given, to be consulted from Statistical refree.
Answer: We completed with Cohen´s d line 389:
Although the effect size between self-reported health and PA is small (Cohen's d = 0.192), it indicates a consistent, positive relationship where individuals who report better health engage in slightly more PA. The effect size for age (Cohen's d = -0.229), and self-reported health (Cohen's d = -.128) and SB indicate that as age increases and self-reported health decreases, SB increases. While this effect size is still small, it is stronger than the effect of self-reported health on PA. For all other indicators, the very small Cohen's d values suggest that the effect of the predictor on the outcome variables is minimal.

Did you not ask body weight in the questionnaire? Which is important when studying PA and SB, it should be mentioned under limitations.

Added in line 359: Another limitation was not including body weight in the analyses. Increments in body weight are described as an alarming effect of lockdown during the COVID-19 pandemic, leading to a potentially higher incidence of overweight, obesity and related health risks [49].

7.Discussion.

Do not write Aim and Hypothesis again under discussion, because these are parts of Introduction.

The aim was to examine PA and SB in a Swedish population before, during, and after

245

the pandemic at four time points: December 2019 (assessed retrospectively), January 2020,

246

January 2022, and December 2022. We hypothesized that the previously reported decrease

247

in PA and increase in SB during the pandemic [26,37] would have recovered to pre-

248

pandemic levels after the pandemic ended.

Removed: The aim was to examine PA and SB in a Swedish population before, during, and after the pandemic at four time points: December 2019 (assessed retrospectively), January 2020, January 2022, and December 2022. We hypothesized that the previously reported decrease in PA and increase in SB during the pandemic [26,38] would have recovered to prepandemic levels after the pandemic ended.

Write the most important finding of your study first then discuss by comparing with other studies.

Discuss about PA and SB among educated vs least educated people, during and after the disease.

Para 3, No need to mention, hypothesis again, go ahead with second most important finding. Discuss the role of Education as determinant of morbidity / Mortality

Removed: Our second hypothesis was that changes in PA and SB during and after the pandemic were associated with age, confirmed COVID-19 infection, education, life satisfaction, occupation, place of birth, self-reported health, and sex.

IHME-CHAIN Collaborators. Effects of education on adult mortality: a global systematic review and meta-analysis. Lancet Public Health 2024; 9 (3): e155-e165.

Singh RB, Singh V, Kulshrestha SK, Singh S, Gupta P, Kumar R, Krishna A, Srivastav SS, Gupta SB, Pella D, Cornelissen G. Social class and all-cause mortality in an urban population of North India. Acta Cardiol 2005; 60 (6): 611-617.

Added and included the above references in line 311: Previous studies show that the years of schooling reduce adult mortality [44] and in an Indian study, members of lower social classes died more often due to infections [45]. However, we have not examined mortality, but our result could be in contrast with those studies.

8.Limitations

Data were self reported by the patients, which is open to bias. The following sentence about participants with Univ. degree were overrepresented, is against your finding given in the Table 1, showing their low number.(Dec 2020, Jan 2022)

Nonmanual workers, participants with a university degree, and participants born in

316

Sweden was overrepresented compared to the overall population [45].

Removed in line 349:
University degree

9.Give few references from 2023 and 2024
Included in line 56: A German study describes that mobility patterns during the nationwide restrictions in the spring of 2020, shifted towards inactivity or low-intensity activities, potentially leading to considerable and lasting health risks [12].
